# Revisiting the distribution of oceanic N$_2$ fixation and estimating diazotrophic contribution to marine production

Weiyi Tang[1], Seaver Wang [1], Debany Fonseca-Batista[2,6], Frank Dehairs[2], Scott Gifford[3], Aridane G. Gonzalez [4,5], Morgane Gallinari[4], Hélène Planquette[4], Géraldine Sarthou[4] & Nicolas Cassar [1,4]

Marine N$_2$ fixation supports a significant portion of oceanic primary production by making N$_2$ bioavailable to planktonic communities, in the process influencing atmosphere-ocean carbon fluxes and our global climate. However, the geographical distribution and controlling factors of marine N$_2$ fixation remain elusive largely due to sparse observations. Here we present unprecedented high-resolution underway N$_2$ fixation estimates across over 6000 kilometers of the western North Atlantic. Unexpectedly, we find increasing N$_2$ fixation rates from the oligotrophic Sargasso Sea to North America coastal waters, driven primarily by cyanobacterial diazotrophs. N$_2$ fixation is best correlated to phosphorus availability and chlorophyll-$a$ concentration. Globally, intense N$_2$ fixation activity in the coastal oceans is validated by a meta-analysis of published observations and we estimate the annual coastal N$_2$ fixation flux to be 16.7 Tg N. This study broadens the biogeography of N$_2$ fixation, highlights the interplay of regulating factors, and reveals thriving diazotrophic communities in coastal waters with potential significance to the global nitrogen and carbon cycles.

[1] Division of Earth and Ocean Sciences, Nicholas School of the Environment, Duke University, Durham, NC 27708, USA. [2] Analytical, Environmental and Geo-Chemistry, Vrije Universiteit Brussel, 1050 Brussels, Belgium. [3] Department of Marine Sciences, University of North Carolina at Chapel Hill, Chapel Hill, NC 27599, USA. [4] Laboratoire des Sciences de l'Environnement Marin (LEMAR), UMR 6539 UBO CNRS IRD IFREMER, Institut Universitaire Européen de la Mer (IUEM), 29280 Brest, France. [5] Instituto de Oceanografía y Cambio Global (IOCAG), Universidad de Las Palmas de Gran Canaria, 35214 Las Palmas, Spain. [6]Present address: Department of Biology, Dalhousie University, Halifax, NS B3H 4R2, Canada. Correspondence and requests for materials should be addressed to N.C. (email: Nicolas.Cassar@duke.edu)

Approximately half of global primary production occurs in the oceans[1]. In the vast expanse of the oligotrophic oceans, marine primary production is limited by nitrogen[2]. In these regions, nitrogen ($N_2$) fixation by diazotrophs has been hypothesized to be an important source of new nitrogen ultimately influencing the uptake and sequestration of $CO_2$[3–5]. For instance, in the North Atlantic subtropical gyre, a significant seasonal carbon drawdown in the absence of measurable nutrients has been attributed to episodic and patchy $N_2$ fixation events[6,7]. These events are, however, difficult to capture using current methods, which rely on discrete sampling. Furthermore, most observations to date have been collected in tropical and subtropical open oceans, overlooking the potential role of coastal regions[8,9]. The limited number of observations impede our ability to close regional marine nitrogen budgets, scale estimates globally, and identify factors controlling $N_2$ fixation[10].

Marine $N_2$ fixation is generally believed to be regulated by various factors including light, temperature, nutrients, and trace-metal availability[11]. While low light, low temperature, and high bioavailable nitrogen have traditionally been assumed to limit $N_2$ fixation, recent studies have reported significant diazotrophic activities in darker, colder, and more nitrogen-rich environments, thereby broadening the putative biogeography of marine $N_2$ fixation[12–16]. The discovery of these new niches has been accompanied by increasing appreciation for the large diversity of species fixing $N_2$, ranging from the well-known *Trichodesmium* and diatom-diazotroph associations (DDA) to more recently-recognized unicellular cyanobacteria[17] and non-cyanobacterial diazotrophs[18].

Taken together, the large uncertainty in regional and global budgets of marine $N_2$ fixation, associated with its patchy and recently broadened biogeography, demands new tools to adequately map this important biogeochemical process. We recently deployed a new method across more than 6000 km of the North Atlantic to revisit the geographical distribution and assess the controlling factors of $N_2$ fixation. Our near-real-time, continuous high-resolution measurements allowed us to locate hotspots of $N_2$ fixation and adapt our sampling strategy to characterize plankton communities and environmental properties[19] (Methods). The contribution of $N_2$ fixation to net community production (NCP) was simultaneously evaluated at high resolution. Our new observations reveal hotspots of marine $N_2$ fixation along the Eastern Seaboard, and highlight the overlooked significance of $N_2$ fixation to coastal and global nitrogen and carbon cycling.

## Results and Discussion

**$N_2$ fixation distribution and controlling factors**. Our survey revealed substantial variability and diel cycling behavior in surface $N_2$ fixation rates, which ranged from less than 0.01 to nearly 15 nmol N $L^{-1}$ $h^{-1}$ (Fig. 1a and Supplementary Fig. 1). When integrated over 24-h $N_2$ fixation diel cycles, continuous estimates of daily surface $N_2$ fixation rates (≤0.19–97.6 nmol N $L^{-1}$ $d^{-1}$) were in line with discrete $N_2$ fixation rates concurrently determined by $^{15}N_2$ incubations ($n = 7$, $r = 0.97$, $p < 0.01$, Fig. 1b and Methods). To extrapolate surface measurements to the entire euphotic zone, we derived an empirical relationship between surface and depth-integrated $N_2$ fixation rates (Supplementary Fig. 2). Our lower-end measurements of less than 10 μmol N $m^{-2}$ $d^{-1}$ are within the range of published rates near Bermuda[20] (Fig. 1c). In contrast, high $N_2$ fixation rates reaching 3000 μmol N $m^{-2}$ $d^{-1}$ near the New Jersey coast are among the top 2% of rates ever reported in the global ocean[21], further underscoring the high $N_2$ fixation along the continental shelf of the eastern seaboard[22], and the value of high-frequency observations for identifying hotspots. The coastal region (bathymetry ≥ −200 m)

stands in sharp contrast to open ocean areas, with depth-integrated $N_2$ fixation rates in the coastal sectors (geometric mean of 577 μmol N $m^{-2}$ $d^{-1}$) being on average an order of magnitude larger than open ocean rates (geometric mean of 85 μmol N $m^{-2}$ $d^{-1}$).

Sea surface temperature did not appear to drive the spatial variability of $N_2$ fixation in our study area (Supplementary Fig. 3 and Supplementary Fig. 4). In open ocean regions, $N_2$ fixation rates varied substantially even when the temperature range was narrow. Moreover, high $N_2$ fixation rates were observed at temperatures ranging from 23 °C off New Jersey to 30 °C near the Florida coast. Fixed nitrogen is known to suppress $N_2$ fixation, but the threshold of inhibition differs among diazotrophs and can be fairly high[23]. Observed dissolved inorganic nitrogen did not effectively regulate $N_2$ fixation rates (Supplementary Fig. 4). We also note that a recent study shows that these coastal waters may be nitrogen limited in summer[24]. In addition, the excess of nitrogen in subsurface waters $\left(N^* = \left[NO_3^-\right] - \left[PO_4^{3-}\right] \times 16\right)$, commonly used as a geochemical proxy for the distribution of $N_2$ fixation[25], was not a strong predictor of overlying $N_2$ fixation rates (Fig. 1c and Supplementary Fig. 4). This comparison should be interpreted with caution, as $N^*$ integrates over longer spatial and temporal scales than our observations. $N^*$ is also not well resolved in coastal waters and may be affected by other processes such as atmospheric nitrogen deposition[26]. In contrast, some of the $N_2$ fixation hotspots coincided with high phosphorus concentration and excess phosphorus $\left(P^* = \left[PO_4^{3-}\right] - \left[NO_3^-\right]/16\right)$ at the ocean surface (Fig. 1b and Supplementary Fig. 4). Regions of $N_2$ fixation have been hypothesized to be coupled to areas of denitrification via the upwelling of waters deficient in nitrogen relative to phosphorus[27]. The phytoplankton bloom near the New Jersey coast, where the highest $N_2$ fixation rates were observed, recurs almost every summer and may be associated with local upwelling[28]. High $N_2$ fixation rates have also been reported in other upwelling systems worldwide, including the equatorial Atlantic Ocean[29], the northwest African coastal upwelling[30], and the Benguela Upwelling System[31]. The excess phosphorus may also result from terrestrial and/or riverine runoff. For example, $N_2$ fixation and carbon sequestration in the tropical North Atlantic were shown to be enhanced by the Amazon River plume[32].

A recent study in the Eastern South Pacific suggested Fe, rather than phosphorus, may limit $N_2$ fixation[33], questioning the spatial coupling between $N_2$ fixation and denitrification[34]. However, we did not find a strong relationship between dissolved Fe and $N_2$ fixation rates across our study (Supplementary Fig. 4). Dissolved Fe ranged from 0.5 nmol $L^{-1}$ near Bermuda to around 1.8 nmol $L^{-1}$ along the Florida coast, which is higher than Fe measured in the Eastern South Pacific. Fe concentration is admittedly a poor predictor of Fe availability, as concentrations merely reflect snapshots of the complex interactions between sources and sinks[35]. Consequently, we calculated $Fe^*$ ($Fe^* = Fe - R_{Fe}PO_4$; where $R_{Fe} = 0.47$ mmol Fe : 1 mol $PO_4$) to evaluate whether Fe potentially limits phytoplankton growth[36]. Positive values of $Fe^*$ across the study area indicate Fe was not limiting. The coupling between $N_2$ fixation and N loss may be a dominant factor in regions where Fe is abundant, notably in coastal oceans. We hypothesize that, in contrast to the Eastern South Pacific, North American coastal waters support substantial $N_2$ fixation due to high sedimentary nitrogen loss[37] and high Fe input (e.g., from sediment and atmospheric dust deposition)[38] (Supplementary Fig. 5). Interestingly, $N_2$ fixation rates correlated well with dissolved manganese (Mn) concentrations. While the correlation of $N_2$ fixation to Mn could be coincidental or symptomatic of other factors, it deserves further investigation as the physiological requirement for Mn in marine diazotrophs is poorly characterized.

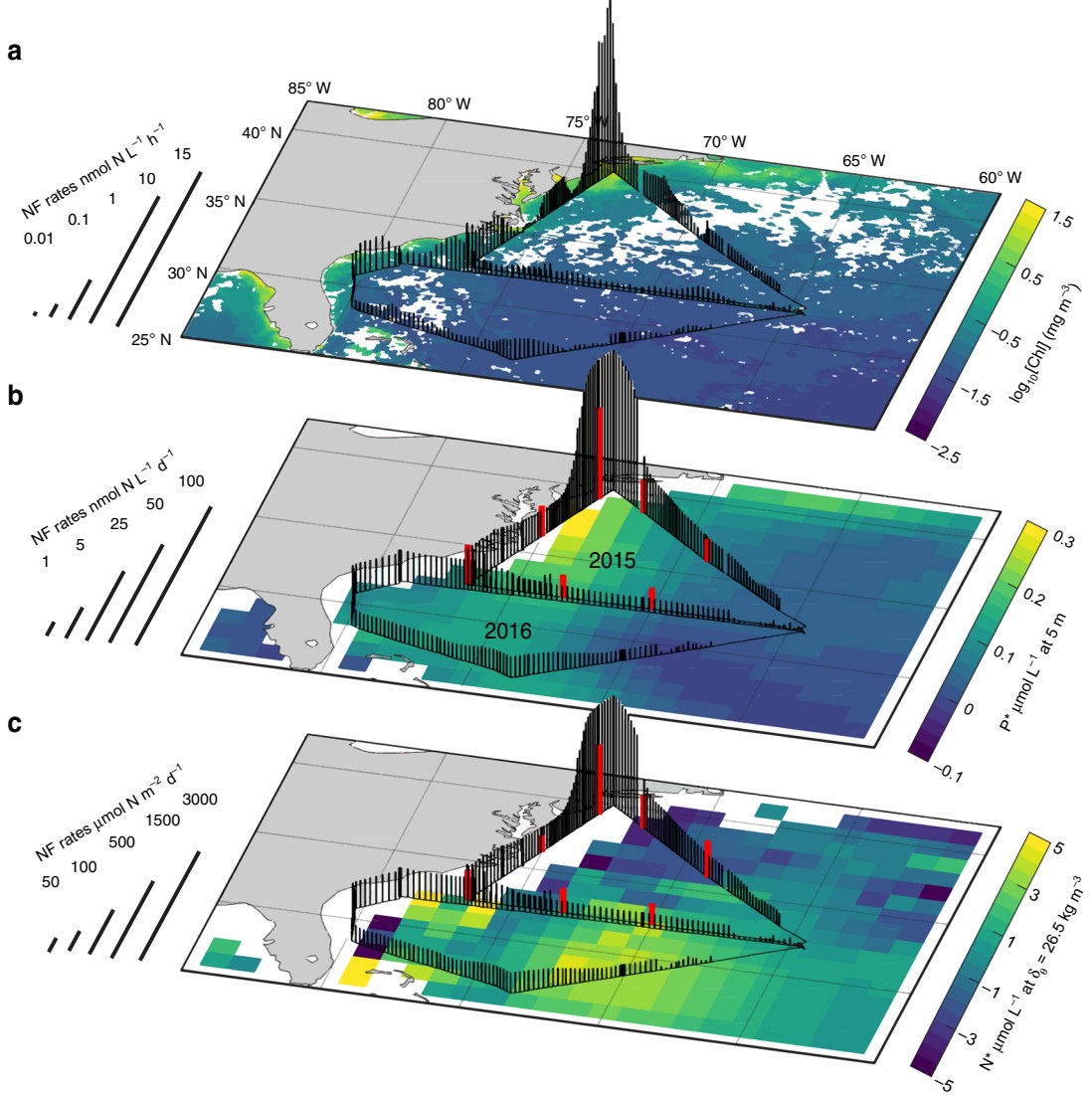

**Fig. 1** $N_2$ fixation rates (NF) measured and calculated in August 2015 and 2016 over the western North Atlantic Ocean. **a** In situ hourly surface volumetric $N_2$ fixation rates determined by the continuous underway incubation method, overlaid on chlorophyll-*a* concentrations measured by the MODIS satellite during the respective cruise periods (Aug 3–12 in both 2015 and 2016). Note that chlorophyll-*a* concentrations are shown on a logarithmic scale. **b** Calculated daily surface volumetric $N_2$ fixation rates overlaid on the August climatology of surface excess phosphate $P^*$ ($P^* = [PO_4^{3-}] - [NO_3^-]/16$). Nutrient data were obtained from the World Ocean Atlas 2013 version 2. Red vertical bars represent $N_2$ fixation rates determined by the discrete dissolved $^{15}N_2$ incubation method during the 2015 cruise. **c** Estimated depth-integrated $N_2$ fixation rates overlaid on the August climatology of subsurface excess nitrogen $N^*$ ($N^* = [NO_3^-] - [PO_4^{3-}] \times 16$)

Our $N_2$ fixation measurements strongly correlated to satellite estimates of chlorophyll-*a* concentrations ([Chl]) (Fig. 1a and Supplementary Fig. 4). This is unexpected as $N_2$ fixation is generally believed to be most significant where nitrogen is limited, such as the low biomass regions of the subtropical gyres. The low-[Chl] waters of the Sargasso Sea, typically viewed as $N_2$ fixation hotspots[39], exhibited lower $N_2$ fixation rates than those measured in the Mid-Atlantic Bight. This pattern was further supported by a meta-analysis, which showed that $N_2$ fixation is correlated to [Chl] in the global ocean (Supplementary Fig. 6). This pattern may be related to the stimulation of non-autotrophic $N_2$ fixation by organic matter in high [Chl] waters[40]. However, while our field observations identify phosphorus, $P^*$, and [Chl] as predictors of spatial variations in $N_2$ fixation in our study area, a meta-analysis of published results shows that none of the putative regulating factors of $N_2$ fixation can satisfactorily explain variations in volumetric rates globally (Supplementary Fig. 6). We posit that

$N_2$ fixation is likely driven by a complex interplay of spatially-variable environmental factors, also reflecting the heterogeneity and the large diversity of marine diazotrophs and their niches.

**Distribution of diazotrophic phylotypes.** Diazotrophs and their potential hosts were identified via high-throughput quantitative 16S rRNA and 18S rRNA gene sequencing from our 2015 cruise (Methods). Although the 16S rRNA gene approach differs from the *nifH* method for characterizing diazotrophs in terms of specificity and coverage[41], it provides some insights into the broad distribution of diazotrophs. To address the cases where organisms may not be capable of $N_2$ fixation despite sharing a similar 16S rRNA gene with diazotrophs, we only searched for diazotrophs known to fix $N_2$ among our 16S rRNA gene sequences. Distinct diazotrophic communities were found to dominate in different ecological domains (Fig. 2a). Heterotrophic groups, which include

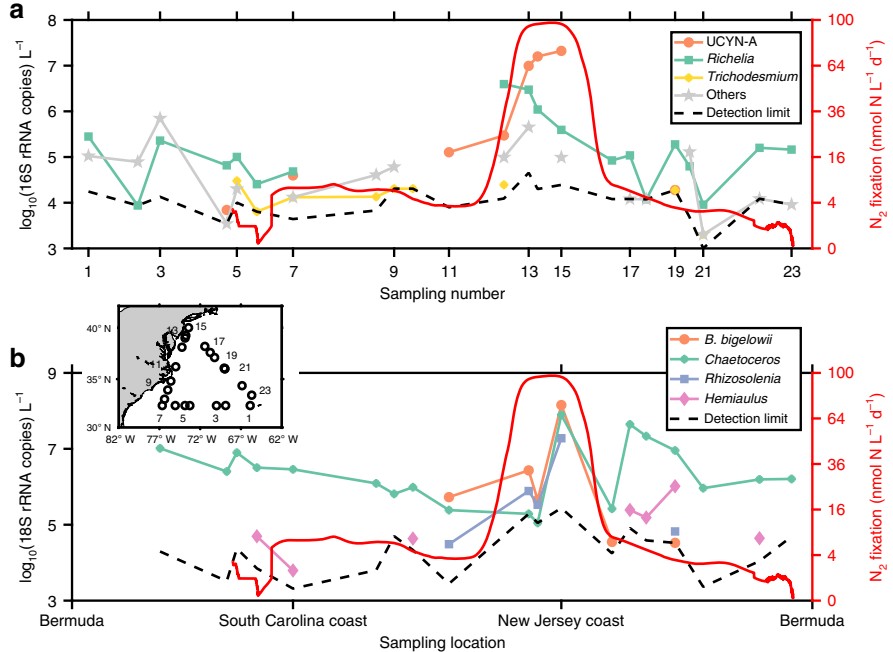

**Fig. 2** Absolute rRNA gene abundances of various diazotrophs and their potential hosts in surface seawater collected during the 2015 cruise. The inset map shows the locations where molecular samples were collected. The distributions of diazotrophs (**a**) and hosts (**b**) are compared with surface daily $N_2$ fixation rates (solid red line). Diazotrophs and hosts below detection limits at different sampling locations are not shown in the figure. "Others" in panel **a** includes diazotrophic proteobacteria belonging to *Bradyrhizobium, Mesorhizobium, Novosphingbium,* and *Paenibacillus*

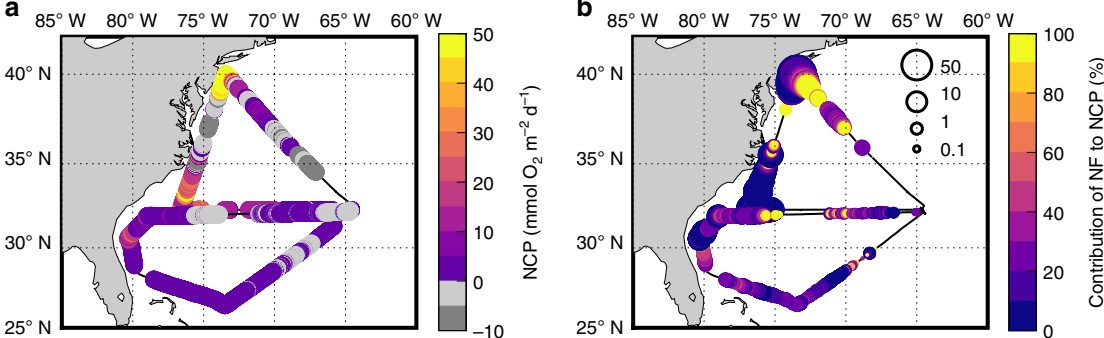

**Fig. 3** Distribution of Net Community Production (NCP) and contribution of $N_2$ fixation to NCP. **a** Color-coded NCP with negative values shown in grey color scale. **b** Distribution of the ratio of $N_2$ fixation-supported carbon production to NCP. Circle size in **b** represents the magnitude of NCP, contextualizing the large uncertainty in calculated ratios when the denominator (i.e., NCP) is small

members known to be diazotrophs[42], were more abundant than diazotrophic cyanobacteria in the open ocean, where $N_2$ fixation was relatively low. These observations are consistent with recent recognition of the widespread distribution of non-cyanobacterial diazotrophs, whose activities remain poorly constrained[18]. *Trichodesmium* peaked off South Carolina. *Richelia intracellularis* showed relatively high abundances in the oligotrophic open ocean and peaked in coastal waters, where its hosts—*Rhizosolenia* and *Hemiaulus*—were also found at relatively high abundances (Fig. 2b). The most striking $N_2$ fixation hotspot off the New Jersey coast was likely driven by a UCYN-A bloom that reached $2 \times 10^7$ 16S rRNA gene copies $L^{-1}$, which is of comparable magnitude to UCYN-A abundances ($2.5–3.5 \times 10^7 L^{-1}$) observed previously in the same region using the *nifH* method[22]. One of UCYN-A's hypothesized hosts, *Braarudosphaera bigelowii* also flourished in this region. Across all samples, the ratio of UCYN-A (16S rRNA gene):*B. bigelowii* (18S rRNA gene) varied from 0.1 to 35 with a median of 0.24. The two organisms co-occurred in most

samples, consistent with previous studies that suggest obligate symbiosis[43,44]. There is a growing interest in UCYN-A's unusual physiological and ecological traits[45]. Its genetic diversity, evolution, and the niches inhabited by its different lineages deserve further investigation. Meanwhile, its presence in the coastal oceans provides new opportunities to study this unique organism. The divergent geographic distribution of different diazotrophs in our research area likely reflects their respective niches. For example, warm seawater is more favorable to *Trichodesmium*[46], while UCYN-A prefers temperate environments[17]. This difference may partly explain why *Trichodesmium* abundances peaked off the South Carolina coast (~30 °C) while UCYN-A dominated in regions with lower temperatures (~23 °C). Overall, the quantitative 16S rRNA and 18S rRNA gene sequencing methods revealed spatial patterns of diazotrophs and their hosts despite the assumptions of our quantitative sequencing methods, e.g., equivalent recovery efficiency for both the standard and the natural sequences in the sample (Methods).

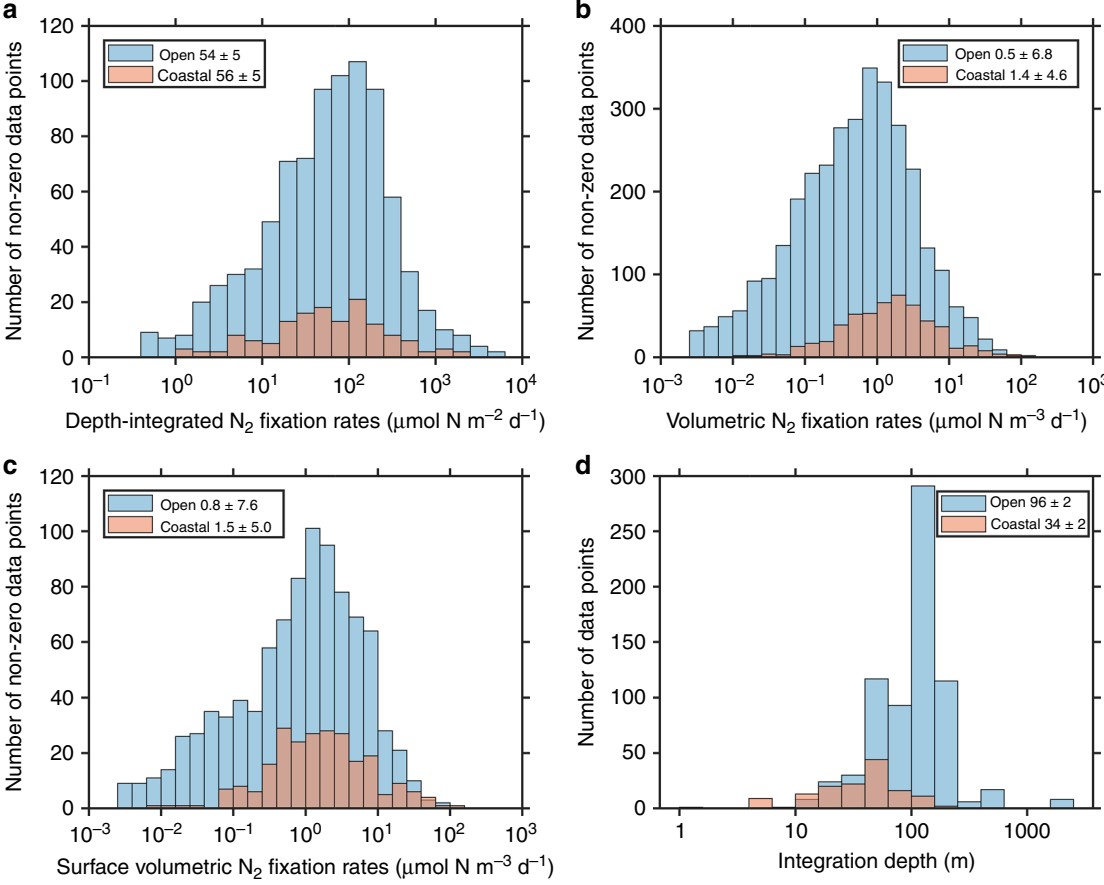

**Fig. 4** Histograms comparing $N_2$ fixation rates in open and coastal oceans. **a** The magnitude of depth-integrated coastal $N_2$ fixation rates (red) is slightly larger than $N_2$ fixation rates in the open oceans (blue). Volumetric $N_2$ fixation rates at all depths (**b**) and at the surface (**c**) are generally higher in coastal regions than in the open ocean. **d** Integration depths used to calculate the depth-integrated $N_2$ fixation rates from the volumetric $N_2$ fixation rates. Geometric means and geometric standard deviations are shown in the legends

**Variable contribution of $N_2$ fixation to new production.** $N_2$ fixation has been estimated to be an important source of new nitrogen in oligotrophic waters, supporting as much as 50% of new production[4,47], yet the contribution of $N_2$ fixation to productivity in coastal oceans remains relatively understudied[48]. To assess the proportion of production fueled by $N_2$ fixation, we combined $N_2$ fixation observations with underway estimates of NCP based on high-frequency measurements of the dissolved $O_2/Ar$ ratio (Methods). NCP was mostly positive, with higher rates along the North American coast where high [Chl] and $N_2$ fixation rates were also observed (Fig. 3a). NCP was relatively low in the open ocean likely due to nutrient limitation. As a rough estimate, we converted $N_2$ fixation rates and NCP to their carbon equivalents using a theoretical C:N:$O_2$ stoichiometry of 106:16:138. Regional differences in stoichiometry would modify but not erase the large gradients of $N_2$ fixation and NCP observed over our cruise transects. We found that the contribution of $N_2$ fixation to NCP varied substantially over the western North Atlantic (Fig. 3b). Across large portions of the oligotrophic subtropical ocean, no more than 20% NCP was generally fueled by $N_2$ fixation (with some high excursions). Other mechanisms of nutrients supply, such as revised estimates of vertical nitrate flux[49], must therefore be invoked to support the NCP rates we observed and those that have been reported in the Sargasso Sea. In contrast, the ratio of $N_2$ fixation to NCP exceeded 50% in some regions off the Cape Hatteras and New Jersey coasts (Fig. 3b). The high contribution of $N_2$ fixation to primary production off the New Jersey coast is supported by dual-tracer $^{15}N_2$ and $^{13}C$ incubations (Supplementary Fig. 7). Despite methodological differences

between these techniques, both methods independently capture similar spatial patterns of contribution of $N_2$ fixation to biological carbon fixation (see NCP estimates in Methods). Our results highlight that $N_2$ fixation is not only high in coastal regions, but may also contribute significantly to marine production.

**Updated $N_2$ fixation budget and global implications.** To contextualize our findings, we performed a meta-analysis combining $^{15}N_2$ incubations collected during our cruises with discrete $N_2$ fixation measurements compiled from the literature, not including our underway continuous measurements (Supplementary Fig. 8). Our updated database contains over 80% more depth-integrated observations (1172 points in total) than the most up-to-date database currently available in the literature (630 points)[21]. Less than 15% of observations reported in the literature were collected in coastal waters. However, these observations support our findings of high $N_2$ fixation rates in the neritic environment (Fig. 4), notably on the eastern American coast[5,22], eastern Arabian Sea[50], and estuaries of the Baltic Sea[14]. The similar magnitude of depth-integrated $N_2$ fixation rates in coastal and open oceans leads to significantly higher volumetric $N_2$ fixation rates in the coastal oceans due to shallower integration depths (Fig. 4 and Methods). However, recent reports of $N_2$ fixation in the deep ocean[12,13] may reverse the pattern of marginally higher depth-integrated $N_2$ fixation rates in coastal waters, if $N_2$ fixation is integrated to the aphotic zone of the open ocean.

In light of these new observations, we reassessed the budgets of marine $N_2$ fixation globally as well as separately for the neritic

**Table 1 N₂ fixation rates in the open and coastal ocean compared with previous studies.**

| Estimates | Measurements | Area (10⁶ km²) | Range of rates[a] (μmol N m⁻² d⁻¹) | Flux (Tg N yr⁻¹) |
|---|---|---|---|---|
| Coastal ocean | 143 | 24.5 | 1.2–397.3[b] | 16.7 (14.3)[b] |
| | | | *1.2–160.2[c]* | *6.6 (0.3)[c]* |
| Open ocean | 857 | 290.2 | 7.5–288.9 | 179.4 (143.4) |
| | | | *7.4–90.9* | *64.2 (2.2)* |
| Global sum | 1000 | 314.7 | 1.2–397.3 | 196.1 (144.1) |
| | | | *1.2–160.2* | *70.8 (2.2)* |
| Luo et al., 2012[21] | 630 | 288.5 | 13–590 | 137 (9.2) |
| | | | *7.9–120* | *62 (52–73)* |
| Großkopf et al., 2012[51] | 335 | 215.1 | 19–115[d] | 177[d] |

[a]Range of mean N₂ fixation rates for different latitudinal bands. Mean N₂ fixation rates in latitudinal bands are shown in Supplementary Table 1
[b]Arithmetic estimates
[c]Geometric estimates in italics
[d]Station weighted estimates. 177 Tg N yr⁻¹ is obtained after correcting for the underestimation due to incomplete dissolution of ¹⁵N₂ in incubations
Uncertainties are shown in parentheses

and oceanic regions (Supplementary Fig. 9). Our updated geometric and arithmetic mean estimates of marine N₂ fixation for the global ocean at 70.8 Tg N yr⁻¹ and 196.1 Tg N yr⁻¹, respectively, are slightly higher than other estimates[21,51] (Table 1 and Supplementary Table 1). Earlier studies report coastal N₂ fixation rates of 15 Tg N yr⁻¹, with most activity associated with benthic diazotrophs[52]. Our updated analysis shows that a significant portion of N₂ fixation is also occurring in the water column of coastal regions, contributing an additional 6.6 (geometric) or 16.7 (arithmetic) Tg N yr⁻¹ to the global budget. This nitrogen input could support the equivalent of 95 Tg C yr⁻¹ of primary production. These updated N₂ fixation budgets have large uncertainties since they are sensitive to the extrapolation method to scale the sparse data to the global ocean. Recent studies have also identified methodological issues in historical observations, which could lead to under-[53] or over-[54] estimations. However, the number of observations resulting from the revised dissolved ¹⁵N₂ incubation method is currently too limited to derive robust global estimates. Nevertheless, after accounting for both aquatic and sedimentary N₂ fixation, the coastal oceans may play a larger role in the nitrogen cycle than previously considered.

Using an underway method recently developed in our lab for continuous high-resolution N₂ fixation measurements, we mapped N₂ fixation at unprecedented scales across the western North Atlantic, identifying hotspots of N₂ fixation in the Mid-Atlantic Bight. Our study challenges the classic paradigm of N₂ fixation distribution and further underscores the central role coastal regions play in the global cycling of nutrients and carbon[55]. With coastal regions being exposed to ever-increasing anthropogenic disturbances[56], expansion of coastal monitoring efforts using high-resolution methods will be critical to evaluate ongoing biogeochemical changes and their global repercussions.

## Methods

**Underway N₂ fixation rate measurements.** N₂ fixation rates were estimated at high resolution using a continuous underway method of Flow-through incubations for Acetylene-Reduction Assays by Cavity ring-down laser Absorption Spectroscopy (FARACAS). A description of the method is presented in Cassar et al. (2018)[19] with a brief outline below. Nitrogenase activity in seawater is estimated based on the conventional technique of acetylene (C₂H₂) reduction to ethylene (C₂H₄)[57–59]. Instead of injecting C₂H₂ gas directly into the incubation bottle, C₂H₂ gas produced from high-purity calcium carbide (Acros Organics) is first dissolved in 0.2-μm filtered seawater that is collected from a trace-metal clean towfish (Geofish) to make a dissolved C₂H₂–H₂O tracer. The dissolved C₂H₂ approach was previously applied for measuring nitrogenase activity in estuarine sediments[60]. The C₂H₂–H₂O tracer is then mixed at a constant ratio using a two-channel peristaltic pump (Masterflex) with a continuous stream of seawater supplied by the Geofish. The mixture of C₂H₂–H₂O tracer and seawater is continuously pumped into a 9-L flow-through incubation reactor (Chemglass) at a flow rate of ~100 mL min⁻¹. The short flow-through incubation, with an e-folding residence time of 90 min (i.e.

~63% of the seawater in the incubation reactor replaced in 90 min), minimizes the effects of C₂H₂ on metabolic processes and on microbial community structure[61,62]. The incubation reactor is lit by a strand of cool-light LEDs fitted with blue filters to simulate the light quality at the ocean surface. The light intensity is instantly calculated and adjusted based on the ship's location and local time. A water jacket on the incubation reactor is flushed with a high-flow rate of continuous surface seawater to mimic the in situ sea surface temperature. Downstream of the flow-through incubation reactor, the seawater flows into a gas extraction chamber. This gas extraction chamber consists of a glass frit with medium-size pores (Chemglass) and a custom-built gas-water separation system. A flow of 35 mL min⁻¹ of C₂H₄-free air controlled by a mass flow controller (OMEGA) continuously purges the incubated seawater, extracting ethylene out of the seawater, and carrying it to a Cavity Ring-Down laser absorption Spectrometer (CRDS, Picarro) for analyses. This CRDS ethylene analyzer measures ethylene concentrations in real time at ppb levels with high frequency and accuracy[63]. Approximately every 3 h, the incubation reactor is bypassed to determine the background ethylene concentration in the mixture of C₂H₂–H₂O tracer and seawater. The difference between the incubation ethylene and background ethylene concentrations represents ethylene production rates during the incubation period. Finally, ethylene production rates are converted to N₂ fixation rates using a conversion factor of 4:1[58,64–66]. We acknowledge that 4:1 is a theoretical ratio with uncertainties. However, our comparison of FARACAS to the ¹⁵N₂ addition method shows good agreement when applying this conversion factor[19]. In the current configuration, the detection limit of FARACAS is 0.19 nmol N L⁻¹ d⁻¹, which is also comparable to the ¹⁵N₂ addition method.

**Discrete N₂ fixation and primary production incubations.** For comparison to our underway survey of N₂ fixation, discrete ¹⁵N₂ incubation experiments in parallel with ¹³C additions were also conducted at eight stations during the 2015 cruise using methods detailed in previous studies[30,51]. Seawater samples were collected from each station at three levels in the euphotic zone, including the surface (5 m), an intermediate depth above the Deep Chlorophyll Maximum (DCM), and the DCM. Four liters of seawater were immediately filtered onto glass microfiber filters (MGF, 0.7 μm, Sartorius) to determine natural concentrations and isotopic signatures of particulate organic carbon (POC) and particulate nitrogen (PN). For incubation experiments, 4.5-L Nalgene polycarbonate bottles were first partly filled with natural seawater. Then, ¹⁵N₂-enriched filtered seawater (98% ¹⁵N atom%, Eurisotop, batch number 23/051301) and NaH¹³CO₃ solution (99%, Eurisotop) were added into the incubation bottles, reaching approximate final enrichments of 2 ¹⁵N atom% and 7 ¹³C atom%. Finally, 4.5-L Nalgene bottles were topped off with natural seawater from sampled depths and capped with septum-fitted screw caps. Incubations were subsequently performed for 24 h in on-deck incubators covered by blue light filters simulating light intensity at the sampled depths. Incubators were flushed with surface seawater to avoid heating due to sunlight. Finally, incubated seawater was filtered onto MGF filters, which were stored at −20 °C until further analysis on land. Filters were treated and analyzed for POC, PN, δ¹³C$_{POC}$, and δ¹⁵N$_{PN}$ using an Elemental Analyzer-Isotope Ratio Mass Spectrometer (EA-IRMS; EuroVector Euro EA 3000 coupled to a Delta V Plus, Thermo Scientific) to calculate corresponding carbon uptake and N₂ fixation rates. The N₂ fixation rates measured by our underway method closely match the results obtained from discrete incubation experiments ($n = 7$, $r = 0.97$, $p < 0.01$). A more comprehensive inter-method comparison can be found in Figure 5 of Cassar et al. (2018)[19], showing good agreement between the two methods.

**Nutrients and trace-metal analyses.** Nutrient samples were collected from a CTD rosette equipped with 24 12-L Niskin bottles. Seawater was subsampled in acid-washed 15-mL polypropylene vials and immediately preserved at −20 °C.

Nitrate + Nitrite and phosphate were analyzed on land using an Automatic Nutrients Analyzer with detection limits of 0.03 μM and 0.014 μM, respectively.

For trace-metal analyses, all reagents, standards, and blanks were prepared in acid-cleaned low-density polyethylene (LDPE) or Teflon-fluorinated ethylene propylene (FEP) bottles. Bottles were cleaned following GEOTRACES protocols. Trace metal samples were collected in surface seawater (~5 m) using a towed fish (UCSC) deployed along side of the ship while underway[67]. During stops, the towfish was recovered from seawater to avoid contamination. Surface seawater was pumped through Teflon tubing to a sink located in a home-made clean plastic bubble installed within the chemistry lab on the ship. There, seawater was filtered in-line from the Teflon tubing outlet using 0.22-μm pore-size Acropak filter cartridges and collected in acid-washed 60-mL LDPE bottles that were triple-rinsed with ~20 mL of filtered seawater before final sample collection. Samples for dissolved trace metal were then acidified to pH = 2 with concentrated HCl (Ultrapur grade, Merck) under a laminar flow hood equipped with HEPA filter. Samples were then double-bagged and stored in the dark, at room temperature, until analysis. On land, all analyses were performed in cleanroom environments at the Pôle Spectrométrie Océans (Brest, France).

Seawater samples were introduced to a PFA-ST nebulizer and a cyclonic spray chamber via a SeaFASTpico introduction system (Elemental Scientific Incorporated, Omaha, NE), following the protocol of Lagerström et al. (2013)[68]. High-purity grade solutions and water (Milli-Q, 18.2 MΩ cm) were used to prepare the following reagents on a daily basis. Buffer was made from 0.5 M acetic acid (Ultrapur grade, Merck) and 0.6 M ammonium hydroxide (Ultrapur grade, Merck) and was adjusted to pH = 8.3. Elution acid was made of 1.6 M HNO$_3$ (Ultrapur grade, Merck) in Milli-Q water and spiked with 1 μg mL$^{-1}$ In (PlasmaCAL calibration standards) to allow for drift correction. Autosampler and column rinsing solutions were made from 0.012 M HCl (Ultrapur grade, Merck) in Milli-Q water.

Mixed element standard solution was prepared gravimetrically using high-purity standards (Fe, Mn, Cd, Co, Zn, Cu, Pb; PlasmaCAL calibration standards) in 0.8 M HNO$_3$ (Ultrapur grade, Merck). A six-point calibration curve was prepared by standard additions of the mixed element standard to our in-house standard (North Atlantic filtered seawater, collected at 55.87445° N/48.09345° W, 40 m depth, 0.15 nM) and run at the beginning, the middle and the end of each run. Final concentrations of samples and procedural blanks were calculated from In-normalized data. Precision was assessed through replicate samples (every tenth sample was a replicate) and accuracy was determined from analysis of consensus seawater (SAFe S1 and D2, and GSP, GSC).

**Diazotrophic community structure analysis**. Diazotrophic phylotypes were identified and quantified using data obtained from 16S rRNA amplicon sequencing of environmental DNA, targeting the V4 region[69]. Eukaryotic hosts of some diazotroph taxa were similarly detected using amplicon sequencing of the V4 region of the 18S rRNA gene. Detailed experimental protocols are described in Wang et al. (2018) including sample collection, addition of internal controls for quantitative sequencing, DNA extraction, primer sequences, PCR amplification steps, and procedures for the analysis of sequencing data[70]. This quantitative analysis has previously been described and applied in other environments[71,72]. Here, the processes are described briefly. From 0.2 to 1 L of seawater (average of 0.8 L) pumped from a towed fish were filtered onto a 0.22-μm polycarbonate filter using a peristaltic pump. The low-volume samples were typically collected in coastal waters, where high biomass led to clogging of filters. The volume filtered was recorded for each sample. The filter was flash-frozen immediately in liquid nitrogen and stored at −80 °C. Following DNA extraction, DNA extraction was performed using the Qiagen DNeasy Plant Mini Kit according to the manufacturer's instructions, with several modifications adapted from Moisander et al. (2008)[73]. Prior to bead beating, 3.04 ng of *Thermus thermophilus* (ATCC #27634D-5) genomic DNA and 0.679 ng of *Schizosaccharomyces pombe* (ATCC #24843D-5) genomic DNA were added to each sample as internal DNA standards, each in 50 μL volumes. These additions introduced ~5,780,000 and 2,800,000 copies of *S. pombe* and *T. thermophilus* rRNA sequences sample$^{-1}$, amounts expected to constitute ≤1% of total reads sample$^{-1}$ following sequencing[70]. PCR cycle parameters are detailed in Wang et al. (2018). Following PCR purification using the Qiagen QIAquick PCR Purification Kit, samples were pooled at equimolar concentrations. Illumina MiSeq sequencing (300 bp PE reads, V3 chemistry) was performed at the Sequencing and Genomic Technologies Shared Resources core facility at the Duke Center for Genome and Computational Biology (Durham, USA). Raw rRNA sequences and metadata are available from the NCBI Sequence Read Archive under accession number SRP126177.

We used QIIME to process and analyze our Illumina sequencing data following the pipeline described in Fadrosh et al. (2014)[74,75]. Taxonomy tables reporting raw counts of 16S rRNA gene and 18S rRNA gene were produced by open-reference operational taxonomic unit (OTU) picking at the 97% threshold using the Usearch 6.1 algorithm and the SILVA ribosomal RNA database[76–78]. The SILVA ribosomal RNA database was supplemented with the addition of full length 16S rRNA gene sequences of UCYN-A1 and UCYN-A2 (accession: NC_013771, CP001842, JPSP01000003, and JPSP01000022). Absolute abundances of the 16S rRNA gene or 18S rRNA gene for each OTU were subsequently calculated based on the number of internal standard sequences recovered[71]. Finally, the concentrations of 16S and 18S rRNA genes in the environment were calibrated for the volume of seawater sample filtered. Common diazotrophs observed from clone library studies across

the global ocean and their eukaryotic hosts were picked out from our 16S and 18S taxonomy tables, respectively[42,73,79–81].

The internal standard method is subject to a number of limitations and caveats[71]. A key assumption of the approach is that the recovery efficiency of the standard is equivalent to the recovery efficiency of the natural sequences in the sample. Variation of rRNA gene copy number is also an important consideration. However, while recovery of the standard may differ from recovery of natural taxa in the same sample, variation in standard recovery efficiency from sample to sample will reflect differences in starting material, losses during elution, and other processes as long as the same PCR and library preparation protocol is followed. In that case, any biases in the quantitative measurement due to amplification biases or DNA extraction recoveries should be consistent across the samples. Therefore, the 16S rRNA approach is informative when providing the spatial distribution and abundance patterns of diazotrophic taxa.

**NCP estimates**. NCP reflects the balance between plankton community photosynthesis and respiration. An excess of photosynthesis leads to a net production of particulate and dissolved organic carbon, which can either accumulate at the ocean surface or be exported to depth. To estimate the proportion of NCP fueled by N$_2$ fixation, we measured NCP underway using the O$_2$/Ar method[82]. Oxygen concentrations in the surface ocean are influenced by biological processes, such as photosynthesis and respiration, as well as physical processes including bubble injection, temperature, and pressure changes. Due to the similar solubility properties of O$_2$ and Ar, the biological O$_2$ supersaturation ([O$_2$]$_{sat}$) can be calculated by removing the effects of physical processes determined from Ar supersaturation ([Ar]$_{sat}$)[83]. Biological O$_2$ supersaturation and undersaturation reflect the metabolic state of the surface ocean, suggesting autotrophic or heterotrophic conditions, respectively[84,85]. Under steady-state conditions within the mixed layer and when vertical mixing is negligible over the residence time of O$_2$ at the ocean surface, NCP can be estimated based on the exchange of biological O$_2$ with the atmosphere using the equations below.

$$NCP \approx k_{O_2} * [O_2]_{sat} * \Delta(O_2/Ar) \qquad (1)$$

$$\Delta(O_2/Ar) = \left[\frac{([O_2]/[Ar])}{([O_2]/[Ar])_{sat}} - 1\right] \qquad (2)$$

$k_{O_2}$ is the gas exchange velocity for oxygen[86,87]. The uncertainties in the NCP estimate are mainly from errors associated with $k_{O_2}$ and vertical mixing of O$_2$. Dissolved O$_2$/Ar ratios in surface seawater were continuously measured by Equilibrator Inlet Mass Spectrometry (EIMS) during the 2015 and 2016 cruises. O$_2$/Ar-NCP estimates were converted to carbon-NCP assuming a constant O$_2$/C stoichiometry[88,89].

We note that our observations of NCP fueled by N$_2$ fixation should be interpreted with caution mainly due to differences in timescales of integration. Our O$_2$/Ar-NCP observations integrate productivity over 3–4 days in this region, while the N$_2$ fixation measurements reflect hourly or daily rates. We cannot rule out the possibility that high N$_2$ fixation rates occurred during late-stages of a phytoplankton bloom when nitrogen was exhausted[29] or, conversely that release of N by diazotrophs relieved N starvation and initiated rapid growth of non-N$_2$ fixers[90]. This artefact in integration timescales is circumvented with dual-tracer $^{15}$N$_2$ and $^{13}$C incubations, which also show a high contribution of N$_2$ fixation to primary production off the coast of New Jersey (Supplementary Fig. 7). In addition, negative NCP values accompanied by detectable N$_2$ fixation and heterotrophic diazotrophs were observed over a large portion of the transition zone between the neritic and open ocean regions. These observations may be attributed to transient net heterotrophy, advective transport of organic matter, or vertical mixing of O$_2$-depleted waters. Further studies within a Lagrangian framework will be required to explore the coupling between N$_2$ fixation and the net metabolic status of marine systems.

The contribution of N$_2$ fixation to NCP measured by FARACAS and O$_2$/Ar method shows a similar spatial pattern as the contribution of N$_2$ fixation to NPP measured by $^{15}$N$_2$/$^{13}$C incubation. $^{13}$C-based primary production measures yield rates closer to NPP than NCP[91]. Therefore, the $^{15}$N$_2$/$^{13}$C-based approach to assessing N$_2$ fixation's contribution to biological production should relate to our FARACAS-O$_2$/Ar according to the following equation:

$$\frac{N_2 \text{ fixation}}{NCP} * \text{export ratio} = \frac{N_2 \text{ fixation}}{NPP} \qquad (3)$$

Where the export ratio = $\frac{NCP}{NPP}$. In some cases, we estimate a contribution of N$_2$ fixation to NCP of 80–100%. Should we assume an export ratio of 8.4% for the oligotrophic Sargasso Sea based on BATS estimates[92], the contribution of N$_2$ fixation to NPP implied by our approach (6–8%) is approximately in line with our discrete incubation-based estimates (Supplementary Fig. 7). Coastal environments likely exhibit higher export ratios of 0.2–0.3[93], which would yield a range of contribution of N$_2$ fixation to NPP of up to 16–30%. Thus, while discrepancies exist between $^{15}$N$_2$/$^{13}$C and FARACAS-O$_2$/Ar-based approaches, the broad relationship between these quantities is as expected.

**Nitrogen budget via $N_2$ fixation in the global ocean**. An updated database of depth-integrated and volumetric $N_2$ fixation rates over the global ocean is presented in Supplementary Fig. 8. The complete dataset of global $N_2$ fixation is shown in the Supplementary Data 1, which includes 1172 depth-integrated and 4299 volumetric $N_2$ fixation measurements. We conducted a Welch's $t$-test to evaluate whether the $N_2$ fixation rates in the coastal oceans (bathymetry $\geq -200$ m) are significantly higher than in the open ocean. This one-tailed hypothesis was examined at the 0.01 significance level. $N_2$ fixation rates were first log-transformed since they are approximately log-normally distributed (Fig. 4a–c). Based on these analyses, the volumetric $N_2$ fixation rates at different depths and at surface are significantly larger in coastal regions than in the open ocean ($p < 0.01$), while depth-integrated $N_2$ fixation rates appear to be similar in both systems.

Nitrogen inputs through $N_2$ fixation were further evaluated for coastal and open oceans separately by scaling to the surface areas of the respective regions. The areal extents of the coastal and open oceans were calculated using ArcGIS. Land (orange), coastal (cyan), and open ocean (blue) regions were delineated using bathymetric contour lines (GEBCO One Minute Grid), with depth criteria of 0 m and −200 m as shown in Supplementary Fig. 9. Surface areas were calculated under a Cylindrical Equal Area Projection (World) with 10° latitudinal bands except at latitudes higher than 50° N/50° S. We calculated global budgets after removing outliers identified using a Tukey's test. These outliers include some extremely high values in the Indian Ocean[50] and some extremely low rates in the eastern North Atlantic Ocean[94] (Supplementary Fig. 8). Global budgets including outliers were also computed but may be substantially biased and are not shown here. Geometric means were used because depth-integrated and volumetric $N_2$ fixation rates are approximately log-normally distributed (Fig. 4a–c). Flux budgets were determined by multiplying the geometric mean of $N_2$ fixation rates by the area of each latitudinal band. Uncertainties were estimated based on the propagation of errors. Regional and global $N_2$ fixation rates are presented in Table 1 and Supplementary Table 1. For comparison, we also present budgets based on the arithmetic mean of $N_2$ fixation rates. Large uncertainties are expected in high latitude regions because of the limited number of observations.

## Data availability

The updated database of depth-integrated and volumetric $N_2$ fixation rates over the global ocean is provided in Supplementary Data 1. In addition, data used in this study are available from the corresponding author on reasonable request.

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

## Acknowledgements

We would like to thank the captains, crew, and marine technicians of the R/V *Atlantic Explorer*.We further thank Geoffrey Smith (UCSC) for his help with deployment of the towfish system. We also thank Zuchuan Li (Duke University, now at WHOI) for providing satellite images for adaptive sampling during the cruises. We are grateful to Yawei Luo (Xiamen University), Run Zhang (Xiamen University), Lasse Riemann (University of Copenhagen), Eyal Rahav (Israel Oceanographic and Limnological Research), Ilana Berman-Frank (Bar Ilan University), Carolin Loscher (Helmholtz Center for Ocean Research Kiel), Camila Fernandez (CNRS), Takuhei Shiozaki (Japan Agency for Marine-Earth Science and Technology), and Ajit Subramaniam (Lamont-Doherty Earth Observatory) for providing access to their datasets. We would like to acknowledge NASA for processing and distributing satellite data on OceanColor Web. This work was funded by an NSF-CAREER grant (#1350710) to N.C. and a Link Foundation Ocean Engineering & Instrumentation Fellowship to W. T. N.C. was also in part supported by the "Laboratoire

d'Excellence" LabexMER (ANR-10-LABX-19) and co-funded by a grant from the French government under the program "Investissements d'Avenir". H.P. was funded by RPDOC BITMAP (ANR-12-PDOC-0025–01). D.F-B. and F.D. were supported by the VUB R&D, Strategic Research Plan "Tracers of Past & Present Global Changes".

## Author contributions

N.C. designed the study. W.T. and N.C. performed the measurements and analysis of $N_2$ fixation and wrote the manuscript with contributions from all co-authors. S.W. performed and analyzed the NCP measurements. S.W., W.T., and S.G. performed molecular sequencing data collection and analyses. D.F.-B. and F.D. performed the $^{15}N_2$ and $^{13}C$ incubation experiments and analyses. A.G.G, H.P., G.S., and M.G. collected and conducted the trace-metal analyses.

## Additional information

**Competing interests:** The authors declare no competing interests.

