## [Peer Review File · Nature Communications]

Reviewers' comments:

Reviewer #1 (Remarks to the Author):

The manuscript 'Coastal oceans broaden the biogeography of marine N₂ fixation' reports N₂ fixation rates using a recently developed continuous measurement of acetylene reduction from two cruises to the Western Atlantic including open oligotrophic and coastal regions. One of the main findings are particularly high rates of N₂ fixation in the more coastal/shelf regions. While measurements in the coastal regions are not entirely new, the application of the modified acetylene reduction leads to an unprecedented resolution of rates across a range of oceanographic conditions. Paired with measurements of O₂/Ar, the presented dataset includes for the first time (I think) a high resolution of N₂ fixation together with an assessment of net community productivity (NCP). This is a very valuable contribution to the N₂ fixation and productivity literature. The manuscript is well written and well structured, and I only have some minor comments below that I hope help the authors to improve their manuscript.

Minor comments:

Title: The title 'Coastal oceans broaden the biogeography of marine N₂ fixation' seems to suggest that this is a new finding while N₂ fixation in coastal regions and in particular in the Western Atlantic has previously been observed. Maybe a different could highlight the new findings of this paper, e.g. the fact "...that N₂ fixation is not only high in coastal regions, but may also contribute significantly to marine production".

Figure 1: I am just wondering whether it has ever been tried to calculate N* and/or P* over an integrated water column in comparison to depth-integrated N₂ fixation and/or primary production rates? Maybe the authors could look into this without the necessity to add something to the manuscript. I'd just be curious.

I 94-96: What about the possibility that the particularly high N₂ fixation rates off the Jersey shore were triggered/supported by freshwater influence? If I look at the salinity values in Supplementary Figure 3, I could imagine that the available P came in through riverine/terrestrial runoff, or a combination of both upwelling and freshwater (is the water column highly stratified in this region). Maybe the authors could comment on this here.

I 111-113: I wonder whether the correlation of N₂ fixation rates with manganese (and probably that of P availability with Mn?) could tell the authors s.th. about the source of the water and the P, upwelling vs. riverine/terrestrial runoff?

I 130 and following: I would appreciate authors could mention the difficulty of matching 16S sequences with nifH (or the potential to fix N₂). There are organisms that share (nearly) identical 16S sequences while one of them is a diazotroph and the other one not (likely the result of horizontal gene transfer). This introduces quite some bias into these analyses that should be mentioned. This is of course easier for cyanobacterial diazotrophs where it is often known whether they can fix N₂, mostly from cultures, or, in the case of UCYN-A and DDAs, from single-cell measurements of field-collected cells. In order to address this issue, maybe the authors could rephrase their sentence stating that they were looking for 'known' diazotrophs in their 16S data. Also, I think it is worth changing the '16S rRNA' to 16S rDNA or 16S rRNA gene (same thing for 18S rRNA) indicating that DNA was sequenced rather than RNA (as far as I can see from the methods).

I 146: Does the analysis here take into account that there are at least two different types of UCYN-A, including the UCYN-A1 clade which is associated to an unknown prymnesiophyte closely related to *B. bigelowii* and UCYN-A2 which appears to be associated to *B. bigelowii*?

I 148: While the Cabello et al. 2015 paper looked at the distribution of the UCYN-A hosts, I think

the obligate symbiosis has been suggested earlier by Thompson et al. (2012, Science).

I 188: I might simply be misunderstanding this sentence. The 80% increase in database sounds like a lot considering that the Luo et al. (2012) database has nearly 3000 entries, but could be true. Do you mean 80% larger with respect to the number of observations, or number of depth-integrated values? Does this include only your data or also rate measurements published elsewhere since 2011/2012? Maybe the authors could add here some information on how many entries have been added to their updated database (it is a lot of work to compile these, particularly if the increase is by 80%), from literature as well as from their cruises.

Supplementary Figure 5 could be turned into a main figure if space is allowed.

Reviewer #2 (Remarks to the Author):

Tang et al. apply their new approach for continuous, highly spatially resolved measurement of ethylene production from acetylene as an index of active nitrogen fixation on two research cruises in the western N. Atlantic in Aug 2015 & 2016. Observations of N₂ fixation were in tandem with concurrent measurements of net community productivity (N₂/Ar ratios), nutrients and trace metal concentrations, and diazotrophic community analysis using 16s rRNA and host 18 sRNA(2015 only). Comparisons are made with traditional discrete tracer assays of N₂ fixation using ¹⁵N₂ and NCP by ¹³CO₂.

The results are impressive. One earlier study (encompassing 3 cruises) by Mulholland et al. 2012 similarly reported substantial rates of N₂ fixation in this region but without the detailed spatial/temporal resolution that the new approach allows. Moreover, the current report documents some extraordinarily high rates of nitrogenase activity at the most northern reaches of their cruise off the coast of New Jersey in the mid-Atlantic Bight, many-fold greater at times than the earlier report.

The author correlate their results with various parameters and find little relationship to temperature, inorganic N or iron concentrations or N*. In contrast, P concentrations and P* did show some correspondence with the spatial distributions of N₂ fixation. Interestingly, dissolved Mn concentrations did positively correlate with N₂ fixation rates observed in the study. There was also a general strong correlation between Chl a and N₂ fixation in the present dataset as well as in the meta-analysis of the global dataset developed in this study.

The authors also provide an updated compendium of rates from studies to date, updating the Luo et al. (2012) summation and provide new global estimates for water column N₂ fixation which incorporate the new findings in coastal waters. The results argue for a much larger contribution for coastal N₂ fixation in the global marine N budget.

Specific comments:

Line 39-40. Admittedly, the numbers are low, but several earlier researchers have considered higher latitude systems- such as Mulholland cited below- but also

Holl, C. M., A. M. Waite, S. Pesant, P. A. Thompson and J. P. Montoya (2007). "Unicellular diazotrophy as a source of nitrogen to Leeuwin Current coastal eddies." *Deep-Sea Research Part II* 54(8-10): 1045-1054.

Needoba, J. A., Rachel A. Foster, Carole Sakamoto, Jonathan P. Zehr and K. S. Johnson (2007). "Nitrogen fixation by unicellular diazotrophic cyanobacteria in the temperate oligotrophic North Pacific Ocean." *Limnology and Oceanography* 52 1317-1327.

Rees, A. P., J. A. Gilbert and B. A. Kelly-Gerreyn (2009). "Nitrogen fixation in the western English Channel (NE Atlantic Ocean)." *Marine Ecology Progress Series* 374: 7-12.

Lines 45-46. Also

Voss, M., P. Croot, K. Lochte, M Mills and I. Peeken (2004). "Patterns of nitrogen fixation along 10°N in the tropical Atlantic." *Geophys. Res. Lett.* 31: 10.1029/2004GL020127.

Bonnet, S., J. Dekaezemacker, T. Moutin, A. N. Knapp, R. Hamersley, O. Grosso and D. G. Capone (2013). "Aphotic N₂ Fixation in the Eastern Tropical South Pacific Ocean." *PLoS One* 8(12):

e81265.

Indeed, some researchers are exploring the Arctic Ocean for this process! (And I do note some Arctic data in the summary).

Lines 129- 158. As the authors appreciate, 16s identification of putative diazotrophs and 18s of hosts is less definitive than a nifH approach. I think it would be worthwhile to discuss the limitations here a bit more than is currently done in the manuscript. I am not a molec. ecologist/ bioinformaticist, but would like to know how (or if) there were means used to more narrowly discern (or to narrow the sequencing results) of likely diazotrophs from closely related non-diazotrophs (e.g. amongst the heterotrophs). Or for the potential diatoms, were the reads specific enough to avoid non-symbiotic forms?

Nonetheless, the UCYN A results are interesting particularly as they align well with the earlier study.

Line 176. I don't see this high % off the Florida coast in Fig. 3b. A few high points on the transect to Bermuda.

Table 1. Could be parsed for consistency in significant figures. Five sf's are way too much implied precision. Similarly for Supp. Table 1.

Line 433. What is an "e-folding" residence time? Explain.

Line 435, Fig. 1 and Supp. Fig. 1. So light intensity in the incubator reflects in situ light relative to time of day? (per Supp. Fig. 1?). The diel patterns reflected in Supp. Fig. 1 does not seem to be reflected in Fig. 1 as clearly.

What are the units on the x axis of Supp Fig. 1, days of the month as per Fig. 1 legend? Perhaps also indicate approx. Sta. # to coordinate with other Figs.

Supp. Fig. 4. I don't see the higher surface values (10-15 nmol N L h) in these plots.

Supp. Table 1, Supp. Fig. 9. Are the far southern zones (> 40° S) omitted because of lack of data?

The authors might wish to note an earlier application of flow thru assay using dissolved acetylene through *Spartina* sediments: Capone, D. G. and E. J. Carpenter (1982). "A perfusion method for assaying microbial activities in estuarine sediments. Applicability to studies of N₂ (C₂H₂) reduction." *Applied Environmental Microbiology* 43: 1400-1405.

Doug Capone

Reply to Reviewers' comments:

We would like to thank both reviewers for their valuable and incisive comments.

Reviewer #1 (Remarks to the Author):

The manuscript 'Coastal oceans broaden the biogeography of marine N₂ fixation' reports N₂ fixation rates using a recently developed continuous measurement of acetylene reduction from two cruises to the Western Atlantic including open oligotrophic and coastal regions. One of the main findings are particularly high rates of N₂ fixation in the more coastal/shelf regions. While measurements in the coastal regions are not entirely new, the application of the modified acetylene reduction leads to an unprecedented resolution of rates across a range of oceanographic conditions. Paired with measurements of O₂/Ar, the presented dataset includes for the first time (I think) a high resolution of N₂ fixation together with an assessment of net community productivity (NCP). This is a very valuable contribution to the N₂ fixation and productivity literature. The manuscript is well written and well structured, and I only have some minor comments below that I hope help the authors to improve their manuscript.

Minor comments:

Title: The title 'Coastal oceans broaden the biogeography of marine N₂ fixation' seems to suggest that this is a new finding while N₂ fixation in coastal regions and in particular in the Western Atlantic has previously been observed. Maybe a different could highlight the new findings of this paper, e.g. the fact "...that N₂ fixation is not only high in coastal regions, but may also contribute significantly to marine production".

In response to the reviewer's comment, we have changed the title to "Revisiting the distribution of oceanic N₂ fixation and estimating diazotrophic contribution to marine production."

Figure 1: I am just wondering whether it has ever been tried to calculate N* and/or P* over an integrated water column in comparison to depth-integrated N₂ fixation and/or primary production rates? Maybe the authors could look into this without the necessity to add something to the manuscript. I'd just be curious.

We calculated the depth-integrated N* (150-400 m) and P* (0-100 m) where the nutrients data are available. The relationships between depth-integrated N₂ fixation and depth-integrated N* and P* are comparable to the ones found for subsurface N* and surface P*

(Figure 1 below). In addition, depth-integrated net primary production rates are overall positively correlated to depth-integrated P*.

Figure 1. N₂ fixation rates versus depth-integrated N* (left) and P* (right).

194-96: What about the possibility that the particularly high N₂ fixation rates off the Jersey shore were triggered/supported by freshwater influence? If I look at the salinity values in Supplementary Figure 3, I could imagine that the available P came in through riverine/terrestrial runoff, or a combination of both upwelling and freshwater (is the water column highly stratified in this region). Maybe the authors could comment on this here.

Overall, we can't ignore the possible input of excess P from runoff, although we don't have direct measurements. Following the reviewer's comment, we added the following sentence in the manuscript "The excess phosphorus may also result from terrestrial and/or riverine runoff. For example, N₂ fixation and carbon sequestration in the tropical North Atlantic were shown to be enhanced by the Amazon River plume (Subramaniam et al., 2008)."

The low salinity (~32 psu) and the shallow mixed layers (~11 m) observed near the New Jersey coast are consistent with fresh water input and some influence from the Labrador Current. While the Hudson River may support high productivity in the coastal region (Moline et al. 2008), the N:P of the Hudson River runoff is generally higher than 16:1 (Howarth et al., 2006 and Lampman et al., 1999) tending to limit the potential for N₂ fixation. Other studies have also highlighted the role of upwelling in supplying nutrients to the summer phytoplankton bloom near the New Jersey coast (Glenn et al., 2004 and Sha et al., 2015).

l 111-113: I wonder whether the correlation of N₂ fixation rates with manganese (and probably that of P availability with Mn?) could tell the authors s.th. about the source of the water and the P, upwelling vs. riverine/terrestrial runoff?

The major sources of manganese in the ocean include lithogenic dust deposition, sediment and rivers (Hulten et al., 2017). Phosphate does not show a clear correlation to Mn (Figure 2). Because the sources and sinks for P and Mn differ, it may be hard to determine the source of the water based on Mn solely.

Figure 2. Phosphate vs Mn in 2016 Bermuda cruise.

l 130 and following: I would appreciate authors could mention the difficulty of matching 16S sequences with *nifH* (or the potential to fix N₂). There are organisms that share (nearly) identical 16S sequences while one of them is a diazotroph and the other one not (likely the result of horizontal gene transfer). This introduces quite some bias into these analyses that should be mentioned. This is of course easier for cyanobacterial diazotrophs where it is often known whether they can fix N₂, mostly from cultures, or, in the case of UCYN-A and DDAs, from single-cell measurements of field-collected cells. In order to address this issue, maybe the authors could rephrase their sentence stating that they were looking for ‘known’ diazotrophs in their 16S data. Also, I think it is worth changing the ‘16S rRNA’ to 16S rDNA or 16S rRNA gene (same thing for 18S rRNA) indicating that DNA was sequenced rather than RNA (as far as I can see from the methods).

Following reviewer’s comments, we modified the sentence to “Although the 16S rRNA gene approach differs from the *nifH* method for characterizing diazotrophs in terms of specificity and coverage (Gaby and Buckley, 2014), it provides some insights into the broad distribution of diazotrophs. To address the cases where organisms may not be capable of N₂ fixation despite sharing a similar 16S rRNA gene with diazotrophs, we only searched for diazotrophs known to fix N₂ among our 16S rRNA gene sequences”.

In addition, we changed “16S rRNA and 18S rRNA sequencing” to “16S rRNA gene and 18S rRNA gene sequencing” to avoid confusion.

I 146: Does the analysis here take into account that there are at least two different types of UCYN-A, including the UCYN-A1 clade which is associated to an unknown prymnesiophyte closely related to *B. bigelowii* and UCYN-A2 which appears to be associated to *B. bigelowii*?

Yes, we have considered both UCYN-A1 and UCYN-A2. Both UCYN-A1 and UCYN-A2 16S rRNA genes were added to our SILVA ribosomal RNA database. We now clarify in the Methods section that “The SILVA ribosomal RNA database was supplemented with the addition of full length 16S rRNA gene sequences of UCYN-A1 and UCYN-A2 (accession: NC_013771, CP001842, JPSP0100003 and JPSP0100022)”. Our UCYN-A sequences are mostly assigned to UCYN-A1.

I 148: While the Cabello et al. 2015 paper looked at the distribution of the UCYN-A hosts, I think the obligate symbiosis has been suggested earlier by Thompson et al. (2012, Science).

In consideration of the reviewer’s comment, we now also cite Thompson et al., 2012.

I 188: I might simply be misunderstanding this sentence. The 80% increase in database sounds like a lot considering that the Luo et al. (2012) database has nearly 3000 entries, but could be true. Do you mean 80% larger with respect to the number of observations, or number of depth-integrated values? Does this include only your data or also rate measurements published elsewhere since 2011/2012? Maybe the authors could add here some information on how many entries have been added to their updated database (it is a lot of work to compile these, particularly if the increase is by 80%), from literature as well as from their cruises.

Following the reviewer’s comments, we now clarify “Our updated database contains over 80% more depth-integrated observations (1172 points in total) than the most up-to-date database currently available in the literature (630 points)”. This calculation and the updated database do not include the new underway observations presented in this study. Rather, the updated database includes both depth-integrated and volumetric N₂ fixation rate measurements published in the literature after 2012. The updated database is provided in Supplementary Dataset 1.

Supplementary Figure 5 could be turned into a main figure if space is allowed.

We prefer to keep this figure in the supplementary material because the various processes putatively influencing the distribution of N₂ fixation as presented in Supplementary Figure 5 are speculative at this stage. However, we will let the Editor and reviewers make the final decision on whether it should be included in the main manuscript.

Reviewer #2 (Remarks to the Author):

Tang et al. apply their new approach for continuous, highly spatially resolved measurement of ethylene production from acetylene as an index of active nitrogen fixation on two research cruises in the western N. Atlantic in Aug 2015 & 2016. Observations of N₂ fixation were in tandem with concurrent measurements of net community productivity (O₂/Ar ratios), nutrients and trace metal concentrations, and diazotrophic community analysis using 16s rRNA and host 18 sRNA (2015 only). Comparisons are made with traditional discrete tracer assays of N₂ fixation using ¹⁵N₂ and NCP by ¹³CO₂.

The results are impressive. One earlier study (encompassing 3 cruises) by Mulholland et al. 2012 similarly reported substantial rates of N₂ fixation in this region but without the detailed spatial/ temporal resolution that the new approach allows. Moreover, the current report documents some extraordinarily high rates of nitrogenase activity at the most northern reaches of their cruise off the coast of New Jersey in the mid-Atlantic Bight, many-fold greater at times than the earlier report.

The author correlate their results with various parameters and find little relationship to temperature, inorganic N or iron concentrations or N*. In contrast, P concentrations and P* did show some correspondence with the spatial distributions of N₂ fixation. Interestingly, dissolved Mn concentrations did positively correlate with N₂ fixation rates observed in the study. There was also a general strong correlation between Chl a and N₂ fixation in the present dataset as well as in the meta-analysis of the global dataset developed in this study.

The authors also provide an updated compendium of rates from studies to date, updating the Luo et al. (2012) summation and provide new global estimates for water column N₂ fixation which incorporate the new findings in coastal waters. The results argue for a much larger contribution for coastal N₂ fixation in the global marine N budget.

Specific comments:

Line 39-40. Admittedly, the numbers are low, but several earlier researchers have considered higher latitude systems- such as Mulholland cited below- but also

Holl, C. M., A. M. Waite, S. Pesant, P. A. Thompson and J. P. Montoya (2007). "Unicellular diazotrophy as a source of nitrogen to Leeuwin Current coastal eddies." *Deep-Sea Research Part II* 54(8-10): 1045-1054.

Needoba, J. A., Rachel A. Foster, Carole Sakamoto, Jonathan P. Zehr and K. S. Johnson (2007). "Nitrogen fixation by unicellular diazotrophic cyanobacteria in the temperate oligotrophic North Pacific Ocean." *Limnology and Oceanography* 52 1317-1327.

Rees, A. P., J. A. Gilbert and B. A. Kelly-Gerreyn (2009). "Nitrogen fixation in the western English Channel (NE Atlantic Ocean)." *Marine Ecology Progress Series* 374: 7-12.

Following the reviewer's comments, we now cite Holl et al., 2007, Needoba et al., 2007 and Rees et al., 2009.

Lines 45-46. Also

Voss, M., P. Croot, K. Lochte, M Mills and I. Peeken (2004). "Patterns of nitrogen fixation along 10°N in the tropical Atlantic." *Geophys. Res. Lett.* 31: 10.1029/2004GL020127.

Bonnet, S., J. Dekaezemacker, T. Moutin, A. N. Knapp, R. Hamersley, O. Grosso and D. G. Capone (2013). "Aphotic N₂ Fixation in the Eastern Tropical South Pacific Ocean." *PLoS One* 8(12): e81265.

Indeed, some researchers are exploring the Arctic Ocean for this process! (And I do note some Arctic data in the summary).

Taking the reviewer's comment into account, we now cite Voss et al., 2004 and Bonnet et al., 2013.

Lines 129- 158. As the authors appreciate, 16s identification of putative diazotrophs and 18s of hosts is less definitive than a *nifH* approach. I think it would be worthwhile to discuss the limitations here a bit more than is currently done in the manuscript. I am not a molec. ecologist/ bioinformaticist, but would like to know how (or if) there were means used to more narrowly discern (or to narrow the sequencing results) of likely diazotrophs from closely related non-diazotrophs (e.g. amongst the heterotrophs). Or for the potential diatoms, were the reads specific enough to avoid non-symbiotic forms?

Nonetheless, the UCYN A results are interesting particularly as they align well with the earlier study.

We used a stringent, commonly-employed criterion of >97% similarity to assign taxonomy to diazotrophs in our 16 rRNA gene sequences. While this approach should discriminate diazotrophs from closely related non-diazotrophs, we plan to examine the differences in identifying diazotrophs using the 16S rRNA gene and the *nifH* gene in a future study. However, following the reviewer's comment, we modified the sentence to "Although the 16S rRNA gene approach differs from the *nifH* method for characterizing diazotrophs in terms of specificity and coverage (Gaby and Buckley, 2014), it provides some insights into the broad distribution of diazotrophs. To address the cases where

organisms may not be capable of N₂ fixation despite sharing a similar 16S rRNA gene with diazotrophs, we only searched for diazotrophs known to fix N₂ among our 16S rRNA gene sequences”.

Line 176. I don't see this high % off the Florida coast in Fig. 3b. A few high points on the transect to Bermuda.

In response to the reviewer's comment, we removed “Florida coast” and clarified “In contrast, the ratio of N₂ fixation to NCP exceeded 50% in some regions off the Cape Hatteras and New Jersey coasts”.

Table 1. Could be parsed for consistency in significant figures. Five sf's are way too much implied precision. Similarly for Supp. Table 1.

The significant figures of numbers presented in Table 1 and Supplementary Table 1 are now modified.

Line 433. What is an “e-folding” residence time? Explain.

We now clarify “with an e-folding residence time of 90 minutes (i.e. ~ 63% of the seawater in incubation reactor replaced in 90 minutes)”.

Line 435, Fig. 1 and Supp. Fig. 1. So light intensity in the incubator reflects in situ light relative to time of day? (per Supp. Fig. 1?). The diel patterns reflected in Supp. Fig. 1 does not seem to be reflected in Fig. 1 as clearly.

What are the units on the x axis of Supp Fig. 1, days of the month as per Fig. 1 legend? Perhaps also indicate approx. Sta. # to coordinate with other Figs.

The light intensity in the incubator is simulated as a function of time of day and the ship location. Supplementary Fig. 1 shows hourly N₂ fixation rates, which is also presented in Figure 1a.

The units on the x-axis of Supplementary Fig. 1 are indeed “days of month in August of 2015 and 2016”. Following the reviewer's comment, the units on x-axis of Supplementary Fig.1 have been clarified.

Supp. Fig. 4. I don't see the higher surface values (10-15 nmol N L h) in these plots.

Log10 transformed surface daily N₂ fixation rates are presented in the correlation analyses shown in Supplementary Fig. 4. This is now clarified in the legend. The units of N₂ fixation rates are now added.

Supp. Table 1, Supp. Fig. 9. Are the far southern zones (> 40S) omitted because of lack of data?

Indeed, only two points are available in the open ocean south of 40°S. No observations are available in the coastal ocean south of 40°S.

The authors might wish to note an earlier application of flow thru assay using dissolved acetylene through *Spartina* sediments: Capone, D. G. and E. J. Carpenter (1982). "A perfusion method for assaying microbial activities in estuarine sediments. Applicability to studies of N₂ (C₂H₂) reduction." *Applied Environmental Microbiology* 43: 1400-1405.

We now cite Capone and Carpenter, 1982 in the Methods session. "The dissolved C₂H₂ was previously applied for measuring nitrogenase activity in estuarine sediments (Capone and Carpenter, 1982)".

References:

Mulholland, M. R., Bernhardt, P. W., Blanco-Garcia, J. L., Mannino, A., Hyde, K., Mondragon, E., ... & Zehr, J. P. (2012). Rates of dinitrogen fixation and the abundance of diazotrophs in North American coastal waters between Cape Hatteras and Georges Bank. *Limnology and Oceanography*, *57*(4), 1067-1083.

Subramaniam, A., Yager, P. L., Carpenter, E. J., Mahaffey, C., Björkman, K., Cooley, S., ... & Capone, D. G. (2008). Amazon River enhances diazotrophy and carbon sequestration in the tropical North Atlantic Ocean. *Proceedings of the National Academy of Sciences*, *105*(30), 10460-10465.

Moline, M. A., Frazer, T. K., Chant, R., Glenn, S., Jacoby, C. A., Reinfelder, J. R., ... & Schofield, O. (2008). Biological responses in a dynamic buoyant river plume. *Oceanography*, *21*(4), 70-89.

Glenn, S., Arnone, R., Bergmann, T., Bissett, W. P., Crowley, M., Cullen, J., ... & Oliver, M. (2004). Biogeochemical impact of summertime coastal upwelling on the New Jersey Shelf. *Journal of Geophysical Research: Oceans*, *109*(C12).

Sha, J., Jo, Y. H., Oliver, M. J., Kohut, J. T., Shatley, M., Liu, W. T., & Yan, X. H. (2015). A case study of large phytoplankton blooms off the New Jersey coast with multi-sensor observations. *Continental Shelf Research*, *107*, 79-91.

Howarth, R. W., Marino, R., Swaney, D. P., & Boyer, E. W. (2006). Wastewater and watershed influences on primary productivity and oxygen dynamics in the lower Hudson River estuary. *The Hudson River Estuary*, 136.

Lampman, G. G., Caraco, N. F., & Cole, J. J. (1999). Spatial and temporal patterns of nutrient concentration and export in the tidal Hudson River. *Estuaries*, *22*(2), 285-296.

Hulten, M. V., Middag, R., Dutay, J. C., Baar, H. D., Roy-Barman, M., Gehlen, M., ... & Sterl, A. (2017). Manganese in the west Atlantic Ocean in the context of the first global ocean circulation model of manganese. *Biogeosciences*, *14*(5), 1123-1152.

Gaby, J. C., & Buckley, D. H. (2014). A comprehensive aligned nifH gene database: a multipurpose tool for studies of nitrogen-fixing bacteria. *Database*, 2014.

Thompson, A. W., Foster, R. A., Krupke, A., Carter, B. J., Musat, N., Vaultot, D., ... & Zehr, J. P. (2012). Unicellular cyanobacterium symbiotic with a single-celled eukaryotic alga. *Science*, *337*(6101), 1546-1550.

Holl, C. M., Waite, A. M., Pesant, S., Thompson, P. A., & Montoya, J. P. (2007). Unicellular diazotrophy as a source of nitrogen to Leeuwin Current coastal eddies. *Deep Sea Research Part II: Topical Studies in Oceanography*, *54*(8-10), 1045-1054.

Rees, A. P., Gilbert, J. A., & Kelly-Gerreyn, B. A. (2009). Nitrogen fixation in the western English Channel (NE Atlantic ocean). *Marine Ecology Progress Series*, 374, 7-12.

Voss, M., Croot, P., Lochte, K., Mills, M., & Peeken, I. (2004). Patterns of nitrogen fixation along 10°N in the tropical Atlantic. *Geophysical Research Letters*, 31(23).

Needoba, J. A., Foster, R. A., Sakamoto, C., Zehr, J. P., & Johnson, K. S. (2007). Nitrogen fixation by unicellular diazotrophic cyanobacteria in the temperate oligotrophic North Pacific Ocean. *Limnology and Oceanography*, 52(4), 1317-1327.

Bonnet, S., Dekaezemacker, J., Turk-Kubo, K. A., Moutin, T., Hamersley, R. M., Grosso, O., ... & Capone, D. G. (2013). Aphotic N₂ fixation in the eastern tropical South Pacific Ocean. *PloS one*, 8(12), e81265.

Capone, D. G., & Carpenter, E. J. (1982). Perfusion method for assaying microbial activities in sediments: applicability to studies of N₂ fixation by C₂H₂ reduction. *Applied and Environmental Microbiology*, 43(6), 1400-1405.

REVIEWERS' COMMENTS:

Reviewer #1 (Remarks to the Author):

My previous comments have been fully addressed and I have no further comments.

Reviewer #2 (Remarks to the Author):

This is a re-review of this contribution which I was favorably inclined and indeed enthusiastic about during the first round of review. It provides a new dimension in spatial coverage of a key process, nitrogen fixation, affecting upper ocean productivity in nutrient poor regions. It also provides unusually high rates of this process in areas where one might not otherwise expect it.

The authors have considered and effectively responded to my earlier comments, incorporating several changes in the manuscript.

Doug Capone